# Direct and Indirect Effects of Essential Oils for Sustainable Crop Protection

**DOI:** 10.3390/plants11162144

**Published:** 2022-08-18

**Authors:** Sabrina Kesraoui, Maria Fe Andrés, Marta Berrocal-Lobo, Serine Soudani, Azucena Gonzalez-Coloma

**Affiliations:** 1Instituto de Ciencias Agrarias, Consejo Superior de Investigaciones Científicas (CSIC), 28006 Madrid, Spain; 2Departamento de Sistemas y Recursos Naturales, E.T.S.I. Montes, Forestal y del Medio Natural, Ciudad Universitaria s/n, 28040 Madrid, Spain

**Keywords:** essential oil, crop protection, direct effects, indirect effects, plant priming

## Abstract

Plant essential oils (EOs) are gaining interest as biopesticides for crop protection. EOs have been recognized as important ingredients of plant protection products including insecticidal, acaricidal, fungicidal, and nematicidal agents. Considering the growing importance of EOs as active ingredients, the domestication and cultivation of Medicinal and Aromatic Plants (MAPs) to produce chemically stable EOs contributes to species conservation, provides the sustainability of production, and decreases the variations in the active ingredients. In addition to these direct effects on plant pests and diseases, EOs can induce plant defenses (priming effects) resulting in better protection. This aspect is of relevance considering that the EU framework aims to achieve the sustainable use of new plant protection products (PPPs), and since 2020, the use of contaminant PPPs has been prohibited. In this paper, we review the most updated information on the direct plant protection effects of EOs, focusing on their modes of action against insects, fungi, and nematodes, as well as the information available on EOs with plant defense priming effects.

## 1. Introduction to Essential Oils 

Essential oils (EOs) are volatile extracts obtained mostly from aromatic and medicinal plants [1]. Lamiaceae is the most important EO-producing plant family and includes about 6900–7200 species belonging to 236 genera distributed all over the world, with the Mediterranean and temperate regions predominating [1]. EOs are an important source of biologically active compounds that have antibacterial, insecticidal, fungicidal, nematicidal, herbicidal, antioxidant, and anti-inflammatory activities [2,3,4]. 

Essential oils are mostly composed of volatile terpenes [2], are usually comprised of 20 to 60 substances, and are characterized in many cases by up to three major components in a relatively high concentration, while other compounds are present in trace amounts [2,3,4,5]. EO components can be grouped into two main groups and four chemical classes (Figure 1): Terpene hydrocarbons, comprising monoterpenes (representing 80% of the EO’s composition) and sesquiterpenes [6,7].Oxygenated compounds, composed mostly of alcohols, phenols, aldehydes, and esters. The aromatic and oxygenated compounds are less abundant than terpenes in EO [5,6,7,8].

Humans have long used medicinal and aromatic plants (MAPs) to produce EOs, mainly as flavors and fragrances based on the traditional use of aromatic plants as culinary herbs and spices [9]. More recently, their use has expanded to include use in human medicine, as phyto-pharmaceuticals, and in aromatherapy [8]. Many of these volatile substances not only have physiological functions but also have multiple ecological functions [10]. They can act as internal messengers, as defenses against herbivores, or as volatiles that also attract pollinating insects to their host.

Lately, the interest in essential oils has focused on their bioactivity acting as biocontrol agents, used at low concentrations against pests and disease-causing organisms, and thus their potential use as alternatives to synthetic chemical pesticides for crop protection and pest control means. This expansion of the potential uses of the EOs has intensified academic and industrial research on the biological activities of these plant compounds [11]. Biocontrol is a widely used concept that has recently gained interest in the impartial control of plant pests [12]. With the increasing recognition of integrated pest management [13], the use of environmental-friendly biopesticides is an excellent alternative to synthetic and contaminant chemicals and molecules [14]. They are biodegradable products with low environmental toxicity and contribute to achieving more sustainable agricultural production methods that meet consumer and societal expectations [15]. The EU framework aims to achieve sustainable use of plant protection products by promoting integrated pest management (Sustainable use of plant protection products—Publications Office of the EU, 2020). In this context, numerous results have described the strong biopesticidal potential of EOs due to their antifungal [16,17], insecticidal [9], or nematicidal activities [18]. However, their direct use as biopesticides has associated problems such as phytotoxicity, which has long been considered a major obstacle to their development into biopesticides (insecticides, fungicides, etc.), their potential impact on the organoleptic quality of the resulting food, and the quantities needed [9]. Additionally, EOs suffer a loss of efficiency when used directly in the field, mainly due to their volatile nature and susceptibility to degradation [17]. Therefore, further research is needed to improve the practical applications of EOs in biocontrol. 

This review presents updated information on Eos’ sustainable production and their direct/indirect plant protection effects. 

## 2. Cultivation and Domestication of MAPs to Produce EOs

The chemical composition of essential oils varies significantly from one region to another [19] and within the same territory depending on the different environmental conditions [20]. The phenological plant phase plays an important role in EO yield and varies with the plant species. EOs from MAPs are usually extracted from plants collected during the flowering period, before the seeds germinate, which may cause a reduction in the regeneration of these plants. Therefore, high yield efficiencies are difficult to obtain, highlighting the importance of the domestication of MAPs [21,22]. 

MAPs are sometimes collected without proper control resulting in loss of habitat. This puts pressure on wild populations of medicinal plants, whose disappearance has accelerated especially in developing countries such as India, China, Nepal, Kenya, Tanzania, and Uganda [23]. Therefore, with the increasing demand for standardized homogeneous raw material in industrial societies, wild MAP species have been domesticated and systematically cultivated for production [24]. Additionally, wild plants useful for humans are being cultivated for conservation as Crop Wild Relatives (CWRs) [25].

Plant domestication offers several advantages over wild harvest for essential oil production. It helps to avoid admixture and adulteration through reliable botanical identification, provides better control over harvested quantities, and facilitates the selection of genotypes with desirable characteristics, especially quality [24]. Additionally, domestication controls the influence on the history of the plant material and postharvest handling [22,23,24,25,26,27]. The key initial steps are MAPs’ exact botanical identification and the detailed description of the growing area. Likewise, it is necessary to carry out a phytochemical evaluation of the initially collected plant material to identify the chemotypes [28]. The domestication process could affect the chemical composition of MAPs and consequently affect their biological activities. For this reason, understanding the influence of domestication is critical to conducting the cultivation of MAP species [29]. The numerous examples of successful MAP domestication include *Origanum* L. *sp*. [29,30,31,32], *Lippia* L. *sp*. [33], *Hyptis suaveolens* (L.) Poit. [34], *Tagetes lucida* Cav. [35], *Artemisia absinthium* L. [36] *Lavandula luisieri* (Rozeira) Rivas Mart. [37], *Mentha* L. *sp.* [38], and *Satureja montana* L. [22].

Additional problems to be considered in MAP crop production are contamination with heavy metals, the damage from pests and diseases, and pesticide residues. Therefore, quality assurance measures are needed to ensure that plants are produced with care so that negative impacts during the wild collection, cultivation, processing, and storage can be limited. The guidelines for good agricultural practices and standards for Sustainable Wild Collection have been established [26,39], including a grazing plan for the conservation of habitats where useful wild species grow [40]. 

## 3. EOs in Plant Protection: Direct Effects

### 3.1. Antifungal Activity

Phytopathogenic fungi are responsible for nearly 30% of all crop diseases and may have a high impact on crops, affecting them during cultivation, postharvest, or storage [17]. There is evidence of EOs’ effects on fungal cells [41,42,43,44,45,46], cell wall alterations [47,48], or gene expression modifications, thus diminishing the fungal virulence [46,49,50].

The effects of EOs against a wide range of fungal species have been extensively studied in vitro but not their mechanisms of action. A recent review on the potential of EOs for the biocontrol of phytopathogens pointed out different mechanisms regarding their antifungal properties [17]. The inhibition of the fungi cell wall formation, by a *Cinnamomum zeylanicum* essential oil has been reported to exert antifungal activity in *Candida albicans*. This essential oil triggered cell cycle arrest by disrupting beta-tubulin distribution, leading to mitotic spindle defects, ultimately compromising the cell membrane, and allowing the leakage of cellular components [51]. Citral, an EO component, inhibited ergosterol biosynthesis in *Penicillium italicum* by affecting the expression of the *ERG6* gene responsible for the synthesis of ergosterol in pathogens [52,53]. The decrease in ergosterol synthesis affected the fungal mitochondria by inhibiting the mitochondrial electron transport. An onion EO, caused the depolarization of mitochondrial membranes by lowering the membrane potential, affecting the ionic Ca++ circuit and ion channels, and the proton pump and ATP pool, decreasing the pH gradient [54]. A decrease in energy metabolism may lead to a slowdown in transcription and translation, consistent with slower ribosome biogenesis and faster RNA degradation [46]. *Melaleuca cajuputi* EO reduced the MIC value (the lowest concentration of drug showing no visible growth) of fluconazole and the expression of *MDR1*, a gene encoding drug efflux pumps in *Candida albicans* [55].

### 3.2. Nematicidal Activity 

Plant-parasitic nematodes are the most destructive group of plant pathogens worldwide, and their control is extremely challenging [56]. Thus, in the last decade, much effort has been focused on the study of the nematicidal activity of EOs and their constituents as potential sources of commercial products for the management of the root-knot nematodes, *Meloidogyne* spp., one of the most economically damaging genera on horticultural and field crops. 

Many EOs extracted from different botanical families have been analyzed in vitro for nematicidal activity mainly against *Meloidogyne spp.* [18]. Among EO-producing plants, some families such as Lamiaceae, Asteraceae, Myrtaceae, Rutaceae, Lauraceae, and Poaceae have been widely studied. Especially the EOs from MAPs of the genera *Artemisia*, *Cympogon*, *Lavandula*, *Mentha*, *Origanum*, *Ocimum*, *Satureja*, *Thymus*, and aromatic trees of the genera *Citrus*, *Eucalyptus*, and *Eugenia*, whose nematicidal effects on root-knot nematodes (*M. arenaria*, *M. chitwoodi*, *M. hapla*, *M. incognita*, and *M. javanica*), have been widely reported [18,57,58,59,60,61,62,63,64].

The mode of action of EOs and their constituents is of practical importance for nematode control because it may provide useful information on the most appropriate formulation and delivery means. The neurotoxic effects on nematodes of EO components have been reported, involving several mechanisms, particularly through GABA, octopamine synapses, and acetylcholinesterase inhibition [18]. Recently, active EOs have been investigated to correlate their mechanism of action for target-specific binding affinities toward the nematode proteins. Molecular modeling and in silico studies suggest a higher binding capacity of geraniol, b-terpineol, citronellal, l-limonene, g-terpinene, α-bulnesene, and α-guaiene to the selected target proteins (ODR1 and ODR3 odorant response genes) and AChE [65,66]. The insight into the biochemical ligand–target protein interactions could be helpful in the selection of biomolecules and essential oils for the development of practically viable bionematicidal products [65].

### 3.3. Insecticidal Activity 

The insecticidal action of EOs has been an area of intensive research lately. According to recent bibliometric analyses, more papers have been published in recent years on this group of natural insecticidal materials than on any other type of chemical class of plant-derived natural products [11,67].

The toxic and sublethal behavioral effects observed in insects and related arthropods can be attributed to the mono- and sesquiterpenoids present in essential oils [9,68]. Low molecular weight terpenoids can inhibit acetylcholineesterase (AChE) enzyme activity in laboratory bioassays, but such bioactivity seldom appears to correlate with toxicity in vivo in target insect species [69]. Recently, a decrease in AChE activity was observed following the exposure of *Alphitobius diapering* to EO of *Illicium verum* and correlated with the loss of refuge-seeking capacity and loss of locomotor capacity [70]. The inhibition of the catalytic activity of AChE, glutathione S-transferase (GST), and catalase (CAT) in *Tribolium castaneum* has been reported for the EOs from *Piper nigrum and Rosmarinus officinalis*. *Citrus sinensis*, *P. aduncum*, and *Zanthoxylum monophyllum* inhibited the GST activity and *L. angustifolia*, *C. sempervirens*, and *Eucalyptus spp.* inhibited the CAT activity [71]. Octopamine receptors have been identified as a target for some of these terpenoids [72,73,74]. Other targets in the insect nervous system include GABA-gated chloride channels [75,76] and the nicotinic acetylcholine receptor [77].

Plant EOs are often complex mixtures of terpenoids, and their bioactivity can be the result of the synergy among constituents [68]. The synergy among the major constituents of rosemary (*R. officinalis*) and lemongrass (*C. citratus*) EOs resulted from increased penetration of toxicants through the insect’s integument [78,79,80,81]. A fumigant experiment against *T. castaneum* with a total of 23 EOs showed that the highest fumigant potential for EOs correlated with a greater diversity in their composition [71]. Furthermore, mixtures in the oil composition can reduce the development of resistance [82] and behavioral habituation to deterrents [83,84]. 

The toxic effects of EOs can upregulate physiologically important proteins and enzymes in insects. *M. arvensis* EO caused high contact toxicity in *Sitophilus granarius* adults and induced dramatic physiological changes in the exposed insects revealed by quantitative proteomics analysis. Most of the differentially expressed proteins (DEPs) were upregulated and related to the development and functioning of the muscular and nervous systems, cellular respiration, protein synthesis, and detoxification [85]. In insecticide-resistant insects, EOs can synergize insecticide toxicity by inhibiting detoxification enzymes. Topical bioassays with the binary mixtures of deltamethrin and individual EOs or their major constituents on the deltamethrin-resistant and -susceptible bed bugs (*Cimex lectularius*) caused a significant increase in their mortality through the inhibition of the P450 activity by EO constituents in resistant bed bugs [86].

## 4. EOs in Plant Protection: Priming Effects 

Several articles have been published characterizing molecular plant responses to inorganic and organic chemicals, plant elicitors, and pathogen or disease-associated molecular patterns (PAMPs and DAMPs respectively). However, a lower number of studies have been conducted to study the contribution of essential oils to combat phytopathogens at the transcriptomic and / or metabolomic level [16]. EOs can act as priming molecules both in biotic and abiotic plant stress responses [87] and are an effective and sustainable tool to control seed-borne diseases [88,89,90]. The metabolomic approach has been used recently for the characterization of metabolic pathways affected by plant priming [87]. 

The priming effects of some EOs improving plant tolerance to biotic stress have been described (Table 1).

Avocado fruit exposed to thyme essential oil had higher antioxidant enzyme activities (SOD, POD, and CAT) than the untreated control fruit [94]. Thyme essential oil was evaluated in its capacity to protect tomato seedlings by the accumulation of peroxidases, which are the first line of defense against ROS [91]. Based on gene expression results, it is postulated that one mechanism by which thyme effectively controls grey mold disease in apple fruit is by inducing the PR -8 gene [90]. 

Mint volatiles triggered conserved signaling pathways in soybean plants and promoted histone modifications of defense genes, including TI and PR1 [100]. The vapor of *O. vulgare* EO triggered a multilayered immune system in grapevine. The analysis of gene expression revealed a complex activation of hormonal interactions involving JA, ET, and SA biosynthesis and their signaling cascades [101]. *Allium sativum* and *R. officinalis* EOs preserved the quality parameters in treated strawberry fruits against *Colletotrichum nymphaeae* due to an increase in their phenolic content and the activity of defense-related enzymes such as peroxidase [98].

Recently, the direct and indirect plant protection effects of *A. absinthium* essential oil on tomato seedlings against *Fusarium oxysporum* have been demonstrated. The EO exhibited an in vitro antifungal effect. In addition, tomato seedlings germinated from seeds pretreated with the EO were protected against the fungus. The EO treatment increased callose deposition and the production of reactive oxygen species (ROS) on seed surfaces and primed a durable defense. The metabolomic analysis of the EO-treated seedlings showed an induction of vanillic acid, coumarin, lycopene, and oleamide in the presence of the pathogen. RNA-seq analysis suggests that AEO treatment could induce de novo epigenetic changes in tomato, modulating the speed and extent of its immune response to *F. oxyxporum*. The EO-seed coating could be a new strategy to prime durable tomato resistance, compatible with other environmentally friendly biopesticides [102].

## 5. Conclusions and Perspectives 

Essential oils are active against plant pests and diseases (fungi, insects, nematodes, etc.), have synergistic effects that lower the risk of resistance development, and therefore are considered ingredients for the development of new biopesticides. However, problems with biomass availability, chemical stability, formulation, and phytotoxicity among others are the major obstacles to their development as biopesticides. Plant domestication and cultivation can partially solve the problems related to chemical stability (chemotypes) and biomass availability, while new formulations can be used to overcome the volatility and stability problems.

Recently, the indirect effects (priming) of EOs on plants have been demonstrated, opening new application opportunities. Most priming studies have been carried out with EO-treated plants and fungal pathogens. Recently, a new application method (seed coating with EO) has shown priming effects in tomato seedlings against a fungal pathogen (*F. oxysporum*) involving metabolic and epigenetic changes in the plant. This application required low amounts of EO and had long-term effects, opening new opportunities for the development of EO-based biopesticides. However, more research is needed to determine the specificity of the plant response to the EO composition and the biotic stress. 

## Figures and Tables

**Figure 1 plants-11-02144-f001:**
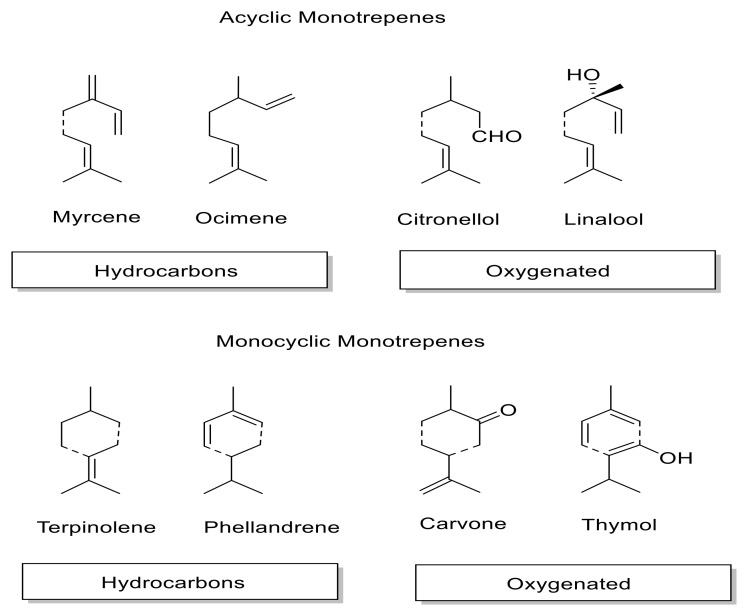
Main chemical classes and selected components found in essential oils.

**Table 1 plants-11-02144-t001:** List of essential oils with reported priming effects.

Phytopathogen	Plant Essential Oil	Primed Plant Crop	Reference
*Botrytis cinerea*	*Satureja hortensis Thymus capitatus* *T. vulgaris*,	Apple, Tomato	[90,91]
*Colletotrichum acutatum*	*Cinnamomum verum*, *Citronella sp.* *Cymbopogon citratus Ocimum basilicum* *T. vulgaris*,	Chili, Mango, Strawberry	[41,92,93]
*C. gloeosporioides*	*C. verum* *S. hortensis* *T. vulgaris* *Zingiber officinale*	Avocado, Mango, Papaya, Pepper fruit	[42,94,95,96]
*C. musae*	*T. vulgaris*	Bananas	[97]
*C. nymphaeae*	*Allium sativum Anethum graveolens Rosmarinus officinalis*	Strawberry	[43,98]
*Fusarium* wilt	*T. capitatus*	Tomato	[91]
*F. solani*	*C. citratus*	Bean	[99]
*Mycosphaerella fijiensis*	*Melaleuca alternifolia*	Bananas	[49]
*Phakopsora pachyrhizi*	*Ment* *ha piperita*	Soybean	[100]
*Plasmopara viticola*	*T. vulgaris* *Origanum vulgare*	Grapevine	[101]

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
