# Peer review of "Direct and Indirect Effects of Essential Oils for Sustainable Crop Protection"

_plants, 2022, doi:10.3390/plants11162144_

Round 1
Reviewer 1 Report
The paper needs some improvement, maybe this papers will help?
https://www.fimek.edu.rs/downloads/casopisi/jatem/issue/v4_2/01-(1)._Acimovic_2021_4(2)_547-557.pdf
https://www.mdpi.com/2077-0472/12/6/874
https://www.fimek.edu.rs/downloads/casopisi/jatem/issue/v4_4/1._Djordjevic_et_al_2021_4(4)_613-618.pdf

Reviewer 2 Report
The authors propose a manuscript titled “Direct and Indirect Effects of Essential Oils for A Sustainable Crop Protection”
The article is original. In particular, this study takes into consideration and highlights an interisting topic on on essential oils frm plants, evaluating their use in different way as active ingredients, the domestication and cultivation of Aromatic and Medicinal Plants in order to produce chemically stable EOs and with the aim to species conservation, sustainability of the production and lower variations in active ingredients. In addition to these direct effects on plant pests and diseases, EOs can induce plant defenses resulting in better protection. The auhors discuss also about the most updated information on direct plant protection effects focusing on their modes of action against insects, fungi and nematodes. I believe that only the introduction and conclusions need to be implemented about the conservation point of view, and in particular the aspects of in situ and ex situ conservation, gene bank, crop wild relatives.
Abstract
In this work instead …In this chapter
1. Introduction
Please complete in the suggested way some statements
· Essential oils (EOs) are volatile extracts obtained mostly from aromatic and medicinal plants [choose reference];
· EOs are an important source of biologically active compounds that have antibacterial, insecticidal, fungicidal, nematicidal, herbicidal, antioxidant, and anti inflammatory activities [2, 3], in the coltivated [choose reference] and wild [Valerio et al. 2021] species.
· Humans have long used medicinal and aromatic plants (MAPs) to produce EOs, mainly as flavors and fragrances based on the traditional use of aromatic plants as culinary herbs and spices [choose reference];
· Many of these volatile substances have not only physiological functions but also play multiple ecological functions [choose reference];
· In this review instead In this review chapter
2. Cultivation and domestication of MAPs to produce EOs
Ø The chemical composition of essential oils varies significantly from one region to another [18], and also within the same territory in relation to the different environmental conditions [Perrino et al. 2021];
Ø EOs from MAPs are usually extracted from plants collected during the flower-ing period, before the seeds germinate, which may cause a reduction in the regeneration of these plants, although the phenological phase of the plant plays an important role and is different from species to species;
· Therefore, with the increasing demand for standardized, homogeneous raw material in industrial societies, numerous wild MAPs species have been domesticated and systematically cultivated [choose reference], as well as already started for the conservation of another groups of wild plants useful for humans as Crop Wild Relatives (CWRs) [Perrino and Wagensommer 2022]
· Domestication offers several advantages over wild harvest for essential oil produc-tion it helps to avoid the admixture and adulteration through reliable botanical identification, gives better control over harvested quantities, and facilitated the selection of genotypes with desirable characteristics, especially quality [choose reference].;
· For botanical point of view and international nomenclature when you cited for the first time in the text the scientific name of genera or species, you must reporting the complete name in the correct way with the name of the author that discovered for the first time the taxon:
Ø Origanum L. sp.
Ø Lippia L. sp.
Ø Hyptis suaveolens L.
Ø T….. lucida please complete name also with genus. T.= ?
Ø Artemisia absinthium …..
Ø Lavandula luisieri …..
Ø Mentha….. sp.
Ø Satureja montana…..
· Guidelines for good agricultural practices and standards for Sustainable Wild Col-lection have been established [22, 35], including a grazing plan for the conservation of habitats where grow the wild species useful for humans live [Perrino et al. 2021]
Add references:
ü Perrino, E.V.; Valerio, F.; Jallali, S.; Trani, A.; Mezzapesa, G.N. Ecological and Biological Properties of Satureja cuneifolia Ten. and Thymus spinulosus Ten.: Two Wild Officinal Species of Conservation Concern in Apulia (Italy). A Preliminary Survey. Plants 2021, 10, 1952. doi: 10.3390/plants10091952
ü Valerio, F.; Mezzapesa, G.N.; Ghannouchi, A.; Mondelli, D. et al. Characterization and Antimicrobial Properties of Essential Oils from Four Wild Taxa of Lamiaceae Family Growing in Apulia. Agronomy 2021, 11, 1431. https://doi.org/10.3390/agronomy11071431
ü Maxted, N.; Ford-Lloyd, B.V.; Jury, S.L.; Kell, S.P.; Scholten, M.A. Towards a definition of a crop wild relative. Biodiversity Conservation, 2006, 15, 2673–2685. https://doi.org/10.1007/ s10531-005-5409-6
ü McCouch, S. ; Baute, G.; Bradeen, J.; Bramel, P.; Bretting, P.; Buckler, E.; Burke, J.; Charest, D.; Cloutier, S., Cole, G.; Dempewolf, H.; Dingkuhn, M.; Feuillet, C.; Gepts, P.; Grattapaglia, D.; Guarino, L.; Jackson, S.; Knapp, S.; Langridge, P.; Zamir, D. Agriculture: feeding the future. Nature 2013, 499, 23–24. https://doi.org/10.1038/499023a
ü Perrino, E.V.; Wagensommer, R.P. Crop Wild Relatives (CWRs) Threatened and Endemic to Italy: Urgent Actions for Protection and Use. Biology 2022, 11, 193. https://doi.org/10.3390/biology11020193.
ü Perrino, E.V.; Musarella, C.M.; Magazzini, P. Management of grazing “buffalo” to preserve habitats by Directive 92/43 EEC in a wetland protected area of the Mediterranean coast: Palude Frattarolo, Apulia, Italy. Euro-Mediterranean Journal for Environmental Integration 2021, 6, 32. https://doi.org/10.1007/s41207-020-00235-2
3. EOs in Plant Protection: direct effects and 4. EOs in plant protection: Priming effects
Well done. No particular observations.
· Remember my suggestion about the correct way on scientific plant name. See my previous comment
Round 2
Reviewer 2 Report
Dear Authors,
I read the work done in the new proposed version. I appreciate that you have followed
all my suggestions with the aim to obtein a more complete work.
The manuscript for my opinion is now complete and
requires no other changes.
Congratulation,
reviewer